# The Dps Protein Protects *Escherichia coli* DNA in the Form of the Trimer

**DOI:** 10.3390/ijms26020619

**Published:** 2025-01-13

**Authors:** Vladislav Kovalenko, Ksenia Tereshkina, Andrey Moiseenko, Yury L. Ryzhykau, Alexander I. Kuklin, Eduard Tereshkin, Petr Zaytsev, Anastasiya Generalova, Nadezhda Persiyantseva, Olga S. Sokolova, Yurii Krupyanskii, Nataliya Loiko

**Affiliations:** 1Semenov Federal Research Center for Chemical Physics, Russian Academy of Sciences, 119991 Moscow, Russia; quebra-mola@yandex.ru (K.T.); ramm@mail.ru (E.T.); vasyazuikova@mail.ru (A.G.); yukrupyanskiy@yandex.ru (Y.K.); 2Faculty of Biology, Lomonosov Moscow State University, 119234 Moscow, Russia; postmoiseenko@gmail.com (A.M.); petrzaytsevrf@gmail.com (P.Z.); sokolova184@gmail.com (O.S.S.); 3Research Center for Molecular Mechanisms of Aging and Age-Related Diseases, Moscow Institute of Physics and Technology, 141700 Dolgoprudny, Russia; rizhikov@phystech.edu (Y.L.R.); alexander.iv.kuklin@gmail.com (A.I.K.); 4Frank Laboratory of Neutron Physics, Joint Institute for Nuclear Research, 141980 Dubna, Russia; 5“N. N. Blokhin National Medical Research Centre of Oncology” of the Health Ministry of Russia, 115478 Moscow, Russia; nadushka99@gmail.com; 6Winogradsky Institute of Microbiology, Research Center of Biotechnology, Russian Academy of Sciences, 119071 Moscow, Russia

**Keywords:** DNA-binding protein Dps, *Escherichia coli*, DNA–Dps crystals, molecular dynamics methods, molecular modeling, SAXS, bacterial survival, oligomeric form of protein, trimer

## Abstract

The Dps protein is the major DNA-binding protein of prokaryotes, which protects DNA during starvation by forming a crystalline complex. The structure of such an intracellular DNA-Dps complex is still unknown. However, the phenomenon of a decrease in the size of the Dps protein from 90 Å to 69–75 Å during the formation of a complex with DNA has been repeatedly observed, and no explanation has been given. In this work, we show that during the formation of intracellular DNA–Dps crystals, the protein transitions to another oligomeric form: from a dodecameric (of 12 monomers), which has an almost spherical shape with a diameter of 90 Å, to a trimeric (of three monomers), which has a shape close to a torus-like structure with a diameter of 70 Å and a height of 40 Å. The trimer model was obtained through the molecular dynamic modeling of the interaction of the three monomers of the Dps protein. Placement of the obtained trimer in the electron density of in vitro DNA–Dps crystal allowed for the determination of the lattice parameters of the studied crystal. This crystal model was in good agreement with the SAXS data obtained from intracellular crystals of 2-day-old *Escherichia coli* cells. The final crystal structure contains a DNA molecule in the through channel of the crystal structure between the Dps trimers. It was discussed that the mechanism of protein transition from one oligomeric form to another in the cell cytoplasm could be regulated by intracellular metabolites and is a simple and flexible mechanism of prokaryotic cell transition from one metabolic state to another.

## 1. Introduction

Dps-like proteins are the major biopolymers produced by prokaryotes, bacteria, and some types of archaea during stationary phase growth to protect their DNA [1,2,3,4,5]. The first such protein was discovered and described in *Escherichia coli* bacteria because of its function in protecting DNA in starving cells [6]. The protein was, therefore, named Dps, which stands for “DNA-binding protein from starved cells” [7]. Further studies have shown that most bacterial genomes contain between one and five Dps genes [8,9] and that the protein has a similar conserved structure in different microorganisms [1].

All Dps proteins also belong to the ferritin-like superfamily [10,11]. The main function of such proteins is to protect the prokaryotic cell from oxidative stress by quenching the toxic interaction between iron in the form of Fe^2+^ and hydrogen peroxide in the Fenton reaction, thereby preventing DNA damage by reactive oxygen species [12,13]. Through the Dps protein, Fe^2+^ is oxidized to Fe^3+^ at specific iron-binding sites in the presence of an oxidant and stored in the protein cavity, which can hold up to 500 Fe^3+^ ions, forming an iron mineral core [14]. Due to their properties and functions, Dps proteins are involved in the protection of cells from various other stress situations: heat and alkali shocks, high pressure, toxic effects of heavy metals, ultraviolet and gamma radiation, multiple stresses during drying, etc. [15,16]. Currently, new functions of this protein are being discovered in bacteria, especially in human and animal pathogens, as this is of great importance for disease prevention [17,18,19].

The structure of the Dps family of proteins has been well studied, especially for *E. coli* Dps. It has been shown that *E. coli* Dps is a dodecamer with 2–3 tetrahedral symmetry, consisting of 12 identical subunits that form spherical particles with an internal cavity of ~4.5 nm. Each subunit of *E. coli* Dps contains 167 amino acids and has a molecular weight of 18.7 kDa [20]. However, the dodecameric form is not the only possible form for this protein. Using electrophoretic fractionation and size exclusion chromatography, dimers, tetramers, and hexamers of Dps have been detected in protein solution [21]. It has also been shown that the dodecameric form of Dps in solution can either be stabilized, e.g., by iron ions [21], or its degradation into smaller oligomeric forms can be induced, e.g., by the sugars D-glucuronate and D-galacturonate (but not D-glucose) [22].

The important DNA-protective function of the protein by forming crystal structures attracted the attention of scientists immediately after its discovery [23,24]. Many interesting discoveries have been made in this area. For example, Dps was shown to interact with the bacterial chromosome via twelve unstructured N-terminal tails containing three lysine residues and one arginine [9]. At the same time, in vitro studies have demonstrated both the ability of Dps to interact non-selectively with different DNA [25] and to show some selectivity, for example, having a higher affinity for branched molecules [26].

However, the most important and still unsolved problem is the packing of DNA in a crystal with Dps in the cytoplasm of bacterial cells. The difficulty lies in the fact that DNA in a crystal does not appear to have an ordered structure and is, therefore, virtually impossible to detect using structural methods [27]. In addition, in vivo studies of living bacterial cells using structural methods have their own additional obstacles [28]. Abraham Minsky et al., who first described the formation of crystalline structures in *E. coli* cells [23] using transmission electron microscopy (TEM) images in which it was impossible to clearly identify the location of the DNA in the crystal, suggested that the DNA was located between tightly packed layers of Dps dodecamers [29]. He also noted that the dimensions of the intracellular crystals are much smaller (68–75 Å) than the Dps dodecamer, which has a diameter of 90 Å, and explained this using the “20% shrinkage of Epon-embedded samples”. However, even the TEM images in the article by the discoverers of Dps, Marta Almiron and co-authors, show that the cell size of the DNA–Dps crystal obtained not in cells but in vitro is 72 Å [6]. At the time of writing, the authors did not know the crystallographic structure of Dps, so they assumed that the change in size was due to the contraction of the protein by the DNA molecule. A similar difference in the size of the crystal lattice of the DNA–Dps crystal in vitro and of the Dps dodecamer itself has been reported by other scientists, who discarded these data as measurement artifacts [28]. We also encountered this phenomenon in our studies, both when examining intracellular crystals in *E. coli* cells [30] and crystals obtained in vitro [31].

However, at the time, we had no explanation for this, so we simply did not focus on it. Now, after analyzing our research and that of other colleagues, we have clarified the situation and propose that the Dps protein changes its oligomeric form when it interacts with DNA, leading to a reduction in the size of the cell in the crystal. This paper presents evidence for the proposed hypothesis and a description of the DNA–Dps crystal model in vivo.

## 2. Results

### 2.1. Study of DNA–Dps Crystal Parameters in Stationary E. coli Top Cells

#### 2.1.1. TEM Study of DNA–Dps Crystal Parameters in Stationary *E. coli* Top Cells

DNA–Dps crystal parameters were studied in samples of 2-day-old stationary cells of *E. coli* Top strain obtained in the normal mode and in the mode of Dps protein overexpression when its content in cells increases 4–6 times from the initial level [30]. Using TEM, bacterial cells with DNA–Dps crystals were detected in bacterial populations in both cases: in the first case, the number of such cells was about (18 ± 4)% of the total number (Figure 1A); in the second case, it was (82 ± 5)% (Figure 1B,C). It should be noted that in bacterial populations of *E. coli* Top without overexpression of the Dps protein, the sizes of DNA–Dps crystals in cells were smaller than in bacteria obtained under conditions of protein overexpression, but the structure of the crystals was identical in both cases.

Analysis of two-dimensional TEM images of more than 40 intracellular crystals in different spatial orientations showed that the periodic structures in them have sizes ranging from 40 to 83 Å (in all three directions), which is significantly smaller than the diameter of the Dps dodecamer (90 Å). This phenomenon of a decrease in the interplanar size of the protein during crystal formation, observed in previous studies, can only be due to a change in its oligomeric form. Modeling the altered quaternary structure of the Dps protein that it acquires in crystals with DNA became the subject of our further study.

In order to determine the parameters of the crystal lattice more precisely, two-axis tomography of the intracellular DNA–Dps crystal was performed, which confirmed that the periodic lattice constants in the tomography plane had a size of 69–75 Å and the periodicity along the tomography planes had values of 40–55 Å (Figure 1D).

#### 2.1.2. Small-Angle X-Ray Scattering (SAXS) of DNA–Dps in Stationary *E. coli* Top Cells to Detect Crystal Parameters

In order to obtain data depending on all six parameters of the crystal unit cell of the studied intracellular crystals, SAXS method, well established in the study of the structure of biological objects, was applied [28]. After prolonged exposure of the cell sample to the X-ray beam, an image of the two-dimensional distribution of scattered radiation intensity was obtained (Figure 2A). It represented a set of concentric rings corresponding to the total scattering from multiple cell crystals in different spatial orientations (powder diffraction) (Figure 2A). By polar angle averaging of the two-dimensional intensity pattern, a one-dimensional intensity curve of X-ray scattering from intracellular crystals as a function of scattering angle 2θ was obtained. This curve contained four distinct peaks whose locations corresponded to the interplanar distances d = 69.6, 54.5, 45.0, and 34.4 Å (Figure 2B).

According to the obtained peak positions, we could accurately determine that the crystal cell cannot place the dodecamer of the Dps protein, which has a diameter of 90 Å. The mismatch of linear lattice parameters in the intracellular crystals (revealed using the SAXS method) confirmed our assumption that the Dps protein in crystals with DNA participates in a different oligomeric form. In order to determine the number of monomers in this oligomeric form, higher-resolution crystal structure data are required. In order to obtain such data, the next stage of studies was carried out on DNA–Dps crystals formed in vitro.

### 2.2. In Vitro Study of DNA–Dps Structure

In vitro, DNA–Dps crystals were obtained under conditions that maximally mimic those found in the cytoplasm of a bacterial cell in the stationary phase of growth. The ring plasmid vector pBlueScript SK+/BaHI 2958 base pairs were used in the experiments, incubated with protein at a DNA–protein mass ratio of 1:25. This DNA–protein ratio was calculated on the basis of known data on the number of protein monomers in the stationary phase of bacterial cells, namely 180,000 Dps dodecamers per cell containing genomic DNA of 4.6 × 10^6^ base pairs in length.

Crystals formed within 3 min of co-incubation of DNA and Dps solutions and were examined using TEM. Images of the crystals in different spatial orientations allowed us to approximate the values of the lattice constants (Figure 3). For a more detailed visualization of the periodic structure of crystals, Fourier filtering of the initial images was applied in the Digital Micrograph program. On the obtained frontal images of the crystal, a layer of approximately identical rounded particles with a diameter of about 7 nm arranged in a hexagonal packing was detected (Figure 3A,B). The side images (perpendicular to the frontal images) showed layers with a periodicity of about 4.4 nm (Figure 3C,D). The crystal thus consisted of toroidal particles with a diameter of 7 nm and a thickness of 4.4 nm, with a central cavity. These values correlate well with those found for intracellular crystals of *E. coli* Top. For comparison, Figure 3E,F shows TEM images of Dps protein in solution with a particle size of ~9 nm.

Given the revealed dimensions of a single toroidal particle in the DNA–Dps crystal ~(70 × 70 × 44 nm^3^), and knowing the size of the protein monomer ~(58 × 25 × 35 nm^3^), it is easy to calculate that no more than four monomers can fit into this oligomeric form. However, taking into account that the toroidal particle has a rather large hole inside, the number of monomers inside is three. This means that this quaternary structure of the protein is a trimer. Apparently, during the formation of the DNA crystal, it initiates the transformation of the oligomeric form of the Dps protein from a dodecamer to a trimer. Molecular modeling techniques were used to determine the spatial structure of this trimer and of the intracellular DNA–Dps crystal itself.

### 2.3. Molecular Modeling of DNA–Dps Trimer

#### 2.3.1. Modeling of Trimeric Structures of Dps Protein and DNA Molecules Relative to Them

To model the trimeric structure of the Dps protein, the dodecameric (the only currently known) crystallographic structure of the Dps protein at atomic resolution (PDB ID 8OUC) was used. The dodecameric structure has two sites of triple contact: (1) a ferritin-like pore and (2) a Dps pore. These two triple monomer junctions were “cut out” in a molecular editor (Figure 4A,B). After separating the trimers from the dodecameric structure, they were relaxed in a solution simulating the salt concentration and pH values in the bacterial cell cytoplasm. The dynamics of the trimeric states of the protein were simulated for 1 μs, and then the Gibbs free energies of the Dps subunit binding into a trimer were calculated using the slow-growth thermodynamics integration (TI) method (see Section 4.7, Section 4.8 and Section 4.9 for details). The trimer formed by the ferritin-like pore (Figure 4A) showed a lower Gibbs free energy of monomer binding ΔΔG_Ferr_ = −185 kJ/mol versus ΔΔG_Dps_ = −115 kJ/mol for the Dps-type pore (Figure 4B). That is, the presence of a subunit in a trimeric state in the form of a ferritin pore is energetically more favorable. Also, visually, the trimer from the ferritin-like pore was more similar to a toroidal structure, so this structure was chosen for further modeling of the DNA–Dps crystal structure.

The layer-by-layer placement of this trimer into the electron density of the crystal obtained from tomography of in vitro crystalline samples was performed using the Chimera program [32]. Initially, the atomic structure of the trimer was manually placed into the electron density of the tomography, observing the alignment of the cavities. Then, the fitting of the atomic structure of the trimer into the electron density was performed based on minimizing the difference between the calculated electron density from the atomic structure of the trimer and the experimental one. A total of 150 trimers were placed, the coordinates of which were analyzed to obtain averaged values of the lattice constants (Figure 4C). Thus, for the coordinates of the trimer centers placed in the electron density of the crystals, the pair distance distribution function and pair angle distribution function were used to identify the most frequently occurring distances and angles between the trimers. Analysis of the distance and angle functions showed that the most frequently occurring distances are 83.3 Å and 54.2 Å, and the angles are 60.5 ° and 121.1°.

These values were satisfied by the space group P1 with the lattice constants a = b = 83.3 ± 3.2 Å, c = 54.2 ± 2.7Å, and α ≅ β ≅ γ ≈ 60.5 ± 4°. Next, the DNA molecule was placed in the electron-dense regions in the through channels of the crystal near the trimers at a distance sufficient for all three Lys N-terminal residues (Lys5, Lys8, and Lys10) of the protein to participate in DNA binding. After placing the DNA molecules in the structure, the trimers were turned with their N-termini closer to the DNA (Figure 4D).

The obtained structure was superimposed with two-dimensional TEM images of DNA–Dps crystals in different orientations (Figure 5). This structure is in good agreement with all crystal projections and explains all the anomalous periodic values detected on two-dimensional TEM images.

#### 2.3.2. Simulation of X-Ray Scattering from a Model Crystal

Based on the obtained structure of the intracellular DNA–Dps crystal, the scattering intensity of X-rays from such crystals in various spatial orientations was simulated. The intensity peaks from the crystal structure were modeled using the FFT program (https://www.ccp4.ac.uk) (Accessed on 10 December 2024) included in the CCP4 v9.0.005 software package. Then, the simulated peaks were blurred using a normal distribution with a standard deviation value corresponding to the peak width from crystals with a linear size of 200 nm. After that, the intensity of all peaks depending on the scattering angle was summed up and presented in Figure 6, together with the experimental intensity from intracellular crystals of 2-day-old *E. coli* Top cells obtained using the SAXS method. As can be seen from the figure, the position of the peaks of the model and the experimental curve are in very good agreement. This indicates the correctness of the constructed DNA–Dps crystal model since the experimental SAXS data give us information about the crystal structure of a very large number of intracellular crystals.

#### 2.3.3. Structure of the DNA–Dps Crystal

The final electron density of the *E. coli* DNA–Dps crystal was obtained by slicing the electron density from the tomography data using the constructed atomic model in the UCSF Chimera program [32]. Only a small number of atoms in the structure lie outside the electron density, indicating good agreement between the constructed model of the DNA–Dps crystal and the experimental data (Figure 7).

## 3. Discussion

The study of the structure of intracellular DNA–Dps crystals is a difficult task requiring good methodological skills in both microbiology and structural biology. This is probably the reason why, apart from the group led by Minsky in 1999–2004, no one has studied the structural features of intracellular DNA stacking in the DNA–Dps crystal of *E. coli* [23,29], despite the widespread interest in various adaptive mechanisms performed by the Dps protein in prokaryotic cells [1,8,9,11,12,13,14]. Biophysicists interested in this problem preferred to study the structure of extracellular DNA–Dps *E. coli* crystals obtained in vitro [27]. However, this approach has significant disadvantages since it is practically impossible to fully recreate in vitro cytoplasmic conditions inside the cell. Therefore, the results obtained in such an experiment may not reflect the real situation occurring during the interaction of nucleoid-associated proteins with DNA. Moreover, there are many factors, both physical and chemical, that influence this interaction [33].

The value of our work consists of the fact that it is based on the “artifact” found in bacterial cells. The paradox of the decrease in the unit cell dimensions of the DNA–Dps crystal compared with the size of the protein dodecamer from which it is supposed to be assembled was first noticed in the works of Minsky et al. and caught our attention [29]. When studying intracellular crystals, we could not find an explanation for this phenomenon for a long time. However, the assumption that the protein changes oligomeric form was the easiest to believe and turned out to be the most realistic. Moreover, the presence of various oligomers (including trimers) of Dps in vivo has been shown in other bacteria, such as *Deinococcus radiodurans* [34] and *Mycobacterium smegmatis* [35].

Thus, having studied the structure of intracellular DNA–Dps crystals in 2-day-old *E. coli* Top cells with and without protein overexpression using TEM in more detail and having made sure once again that the unit cell size of the crystal is no more than 75 Å, we were able to continue the in vitro experiments and substantiate the “trimeric” model of the intracellular DNA–Dps crystal. In addition, the study of the *E. coli* Dps protein and its interaction with DNA using classical molecular dynamics methods in the all-atom approximation made a great contribution to the understanding of the processes taking place. Thus, the study of the mechanism of subunit interaction in the dodecamer of the Dps protein showed that each subunit is in direct contact with five others (Figure 8A, attention to the subunit colored green). Moreover, two contacts were associated with ferritin-type pores (Figure 8B), and two contacts were associated with Dps-type pores (Figure 8C). However, the potential interaction energy was lower for the subunits constituting the ferritin-like pore (Figure 8D), having also a larger contact area (Figure 8E–H). These simulations helped to establish a further vector of research related to the selection of the trimer formed by the ferritin-like pore.

As a result, the DNA–Dps crystal model constructed from the experimental TEM data and using molecular modeling techniques showed good agreement with the data obtained in the SAXS experiment from 2-day-old cells, which confirmed the validity of our conclusions.

The use of three approaches—in vivo, in vitro, and in silico—led to a successful deciphering of the mechanism. It should be noted separately that, working with bacteria of different *E. coli* strains over a long period of time, in different physiological states (different resting periods), and under different stresses (temperature, oxidative, starvation, etc.), we observed the formation of DNA–Dps crystals with the same cell size of 69–75 Å in the cytoplasm of cells. This suggests that such a crystal structure, in which the protein is in the form of a trimer, is a typical situation for a bacterial cell. It is, therefore, likely that the cell has well-established mechanisms for transferring Dps from one oligomeric form to another, using the structural dynamic plasticity of the protein. The dodecameric form of the protein is required for prokaryotic cells to scavenge and store iron ions to protect against oxidative stress during active growth [12,13]. On the other hand, as the bacteria approach the stationary phase of growth, the metabolic activity of the bacteria decreases, and a large amount of Dps protein is synthesized. This reduces the iron/protein ratio, destabilizing the dodecameric form [21]. Other extracellular metabolites, such as some sugars, may also be involved in the process of “collapsing” the dodecameric form [22]. It is also possible that the oligomeric transition is associated with an effect (blocking) on the N-terminus of the Dps protein, as has been shown for *M. smegmatis* [36,37]. Thus, as a result of certain mechanisms, the oligomeric form of the *E. coli* Dps protein is modified to perform other functions, in particular to interact with DNA for its protection. Scientists studying oligomeric forms of Dps from *D. radiodurans* came to the same conclusion about the existence of a regulatory mechanism that modulates the oligomeric equilibrium and depends on growth stages and environmental conditions [34].

Molecular dynamics modeling suggests that Dps trimers have multiple internal interactions to be monolithic functional units. In this case, the N-terminal regions of the protein can both strengthen the trimeric state of the protein by participating in the binding of subunits to each other and form bonds with DNA by easily moving into the region between neighboring trimers. The main subunit binding sites in the trimer are amino acid residues such as Glu82-Lys157 (Figure 9A), Arg83-Asp156 (Figure 9B), Asp143-Arg153 (Figure 9C), and Asp20-Arg133 (Figure 9D). The positively charged Arg18 at the flexible N-terminus has multiple contacts with negative regions of the protein defined by the amino acid residues Glu120, Asp123, Asp131, and Glu138 (Figure 9E). Modeling with a double-stranded DNA molecule (5‘-GTACTATATATTATTATGGGGGGTGATGGATGGATGGATA-3’) showed that the flexible N-termini have the ability to bind DNA (Figure 9F). Interestingly, arginine residues are predominantly responsible for subunit binding within the protein ferritin-like trimers, while lysine residues are responsible for DNA binding.

Studying the mechanisms of intracellular transition of the dodecameric form of Dps protein into trimeric form and back in interaction with DNA is an important microbiological task, as it is directly related to the ability of bacteria to quickly switch to a long-term quiescent state under unfavorable conditions and also quickly restore viability when a favorable period occurs. Possessing the levers of such transitions, humans could use them to solve key problems in medicine and biotechnology. Also, the study of these mechanisms opens new perspectives for the use of Dps protein as a compartment for targeted drug delivery [38], allowing us to discover conditions when drug loading occurs in the cavity of the dodecameric protein and unloading at the right place during a controlled transition to the trimeric oligoform. Further studies will make it possible to study the discovered biological phenomenon in more detail and to understand the mechanisms of its regulation.

## 4. Materials and Methods

### 4.1. Bacteria and Cultivation

The objects of the study were the Gram-negative bacteria *Escherichia coli* Top10/pBAD-Dps (hereinafter *E. coli* Top) from the Biotechnology Research Center collection [30]. These strains are genetic constructs that contain plasmids containing a DNA region encoding the Dps protein, allowing for the production of cells with overproduction of Dps protein.

Bacteria were grown in Luria-Bertani (LB) (Broth, Miller, VWR, Radnor, PA, USA) medium with the addition of 150 µg/mL ampicillin. The inoculum, a steady-state growth phase culture (overnight culture), was added in an amount of 1 mL per 50 mL of medium. Cultivation was carried out in 250 mL glass flasks with cotton plugs and 50 mL of nutrient medium under stirring (140 rpm) at 28 °C for 2 days. To obtain bacteria in which the amount of Dps protein was significantly higher than normal, protein expression was induced by adding 6.7 mM arabinose to cultures of strains of the linear growth phase.

### 4.2. Sample Preparation for Electron Microscopy

Cells were fixed with 2% glutaraldehyde for 5 h and, after re-fixation with 0.5% paraformaldehyde, washed with 0.1 M cacodylate buffer (pH 7.4), counterstained with 1% OsO4 in cacodylate buffer (pH = 7.4), dehydrated in an increasing series of ethanol solutions followed by desiccation with acetone, impregnated, and embedded in Epon-812 (according to the manufacturer’s instructions). Ultrathin sections (70–200 nm thick) were cut with a diamond knife (diatom) on an Ultracut-UCT ultramicrotome (Leica Microsystems, Wetzlar, Germany), transferred to 200 mesh copper grids coated with Formvar (SPI, Lakewood, WA, USA), and counterstained with lead citrate, according to the established Reynolds procedure.

### 4.3. Transmission Electron Microscopy (TEM) and Tomography of Cell Samples

Ultrathin cell sections were examined in a JEM-2100 transmission electron microscope (JEOL, Tokyo, Japan) with an accelerating voltage of 200 kV and a magnification of ×13,000–21,000. Images were recorded using Ultrascan 1000XP and ES500W CCD cameras (Gatan, Pleasanton, CA, USA). Tomograms were obtained from 100–200 nm thick sections using JEOL tomography software (version 4.9.10) (JEOL, Tokyo, Japan). The goniometer tilt angle ranged from −60 to +60 (with a constant step of 1 degree). The image series were aligned using Digital Micrograph software (version 1.5.46) (Gatan, Pleasanton, CA, USA) and then reconstructed using the back projection algorithm in IMOD4.11. Three-dimensional subtomograms were visualized in the UCSF Chimera package version 1.13 [32].

### 4.4. SAXS of Cell Samples

The cellular mass was placed into PCR-tubes using a syringe. The tubes were then sealed with Parafilm M, followed by SAXS measurements of the samples under vacuum. SAXS measurements were performed using the instrument with rotating anode generator Rigaku MicroMax-007HF (Rigaku, Tokyo, Japan) at MIPT (Dolgoprudny, Russia) previously used and described in [39]. The X-ray patterns were measured using a 2D position-sensitive multiwire gas-filled detector Rigaku ASM DTR Triton 200 (Rigaku, Tokyo, Japan) placed at a distance of 2.0 m from the sample; exposure time was 4 h for each sample. The scattering patterns were radially averaged to obtain the scattering intensity as a function of the scattering vector q = 4π/λ sin(θ), where λ = 0.1542 nm is the X-ray wavelength, and 2θ is the scattering angle.

### 4.5. TEM of Extracellular DNA–Dps Crystals

For TEM studies, 3 μL of purified Dps (3.4 mg/mL) were added to 1.5 μL of DNA (the ring plasmid vector pBlueScript SK+/BaHI 2958 base pairs at a concentration of 1.04 μg/mL) directly into a carbon-coated copper TEM grid (SPI) followed by the immediate addition of 1.5 μL of EDTA (0.14 mM). In this way, crystals were formed directly on the TEM grid. After incubation for 15 s, excess liquid was quickly removed with filter paper, and each grid was stained with 1% aqueous uranyl acetate for 60 s and air-dried after stain removal.

Grids were examined on a JEM-2100 analytical transmission electron microscope (JEOL, Tokyo, Japan) equipped with a LaB_6_ filament with an accelerating voltage of 200 kV. Images were acquired using an Ultrascan 1000XP CCD camera (Gatan, Pleasanton, CA, USA) under low dose conditions with a defocus of 0.5–1.0 mm.

### 4.6. Model Building

Three-dimensional models of Dps trimers were constructed in the UCSF Chimera software package version 1.13 from the dodecameric structure of the protein, solved using X-ray crystallography (PDB ID 8OUC). The initial positions of the three monomers that make up the trimers were obtained by cutting out three subunits of the protein, namely, those forming the ferritin-like pore and the Dps-type pore. The Dps dodecamer was built based on the same model. Also, using UCSF Chimera, the N-termini of the protein, missing from the structural file, were manually completed. The double-stranded DNA 5′-GTACTATATTATGGGGTGATGGATA-3′ of 25 base pairs in B-form was modeled in UCSF Chimera.

### 4.7. All-Atom Molecular Dynamics of Dps Dodecamer

In order to determine the mechanisms of interaction of subunits in the protein dodecamer, the dynamics of the protein in water were investigated. The studies were carried out in the Gromacs 5.1 [40] package using an all-atom force field AMBER99-PARMBSC1 [41] and an SPC/E water model. MD protocol included potential energy minimization using the steepest descent method, followed by relaxation of the system for 200 ps at constant volume and then pressure. During the simulation, the temperature of 310 K was maintained using a Langevin thermostat [42] with a friction constant of 0.5 ps^−1^, and a pressure of 1 atm was maintained using a Parrinello-Rahman barostat [43] with a time constant of 2 ps. Electrostatic interactions over long distances were calculated using the Ewald summation method (PME). The cutoff radii for all types of interaction were taken to be equal to 1.5 nm. The simulation time was 0.5 μs.

### 4.8. Coarse-Grained Molecular Dynamics of DNA–Dps Crystals

The molecular dynamics simulations were performed using Gromacs 5.1 [40]. To speed up the simulations, the coarse-grained MARTINI 2.2 force field [44] was chosen. The periodic box contained a Dps trimer (1086 particles), sodium (203), chlorine (193), and calcium (1) particles, as well as 25,385 water particles, 10% of which were modeled using “antifreeze” water particles to avoid artifactual freezing [45]. The difference between “antifreeze” water (WF) and “ordinary” water (W) is as follows. Non-bonded interactions of MARTINI particles are described using the shifted Lennard-Jones 12–6 potential energy function:(1)$$U_{LJ}\left(r\right)={4\varepsilon }_{ij}{[\left(\frac{\sigma_{ij}}{r}\right)}^{12}-{\left(\frac{\sigma_{ij}}{r}\right)}^{6}],$$
where $\varepsilon_{ij}\;$ is the strength of their interaction between the particles *i* and *j*, and $\sigma_{ij}$ is the closest distance of approach between them. When interacting with all atoms of the system and particles of its type, WF particles behave like molecules of W: for example, *σ*(W − W) = 0.47 nm, *σ*(WF − WF) = 0.47 nm. However, the distance of W and WF was increased so that *σ*(W − WF) = 0.57 nm, which allows to avoid freezing.

The systems were carried out through the procedure of energy minimization by the steepest descent method and two-stage relaxation at a constant number of particles, temperature, volume (NVT ensemble, 0.2 ns), and then pressure (NPT ensemble, 0.2 ns).

The integration step was 10 fs. To maintain the temperature of 310 K, a Langevin velocity thermostat [42] with a time constant of 0.5 ps was used. The cutoff radii for the Coulomb and van der Waals interactions were 1.2 nm. A Parrinello-Rahman barostat [43] with a time constant of 4 ps provided an isotropic pressure of 1 bar, and the isothermal compressibility of water was taken to be 4.5 × 10^−5^ bar^−1^. The LINCS algorithm was used to constrain the fast degrees of freedom. The systems were simulated within 1 μs.

### 4.9. Thermodynamic Integration

After the classical molecular dynamics simulations, the free energy of binding of protein monomers in both types of trimers was studied. For this purpose, the obtained trimer structures and the protein monomer simulated according to the above molecular dynamics protocol were subjected to computations using the slow-growth thermodynamic integration method. This method has proven itself well and has been updated for biomolecules [46,47]. In this work, it was used according to our previously developed protocol for calculating the free energy of DNA binding to the Dps protein [48]. The simulation was performed using 72 λ-points of the slow growth method, 20 ns for each point, in the NPT ensemble. The following Gibbs free energy values were obtained. The solvation energy of a protein monomer, i.e., the transition of a protein subunit from the “ghost” state to the fully solvated one, ΔG_1_ = −1787.7 ± 27.7 kJ/mol. The energy of transition of protein subunit from the “ghost” state to the bound state in Dps-type trimer ΔG_2_ = −1903.16 ± 32.05 kJ/mol. The energy of transition of protein subunit from the “ghost” state to the bound state in ferritin-like trimer ΔG_3_ = −1973.68 ± 27.08. The final binding energies of subunit in each type of trimer were calculated as the differences ΔΔG_Dps_ = ΔG2 − ΔG1 and ΔΔG_Ferr_ = ΔG3 − ΔG1. This allowed us to determine the following rounded Gibbs free energy values: ΔΔG_Ferr_ = −185 kJ/mol for the ferritin-like pore and ΔΔG_Dps_ = −115 kJ/mol for the Dps-type pore.

## 5. Conclusions

The Dps protein protects the DNA of the bacterial cell *E. coli* under starvation conditions by forming a crystalline complex with it, the structure of which is still not completely known. The Dps protein was thought to participate in the formation of this complex only as a 90 Å dodecamer. Our study refuted this assertion. By combining three approaches, in vivo, in vitro, and in silico, we have shown for the first time that in the intracellular DNA–Dps complex of *E. coli*, the protein is in the form of a trimer with a size of ~(70 × 70 × 44) nm^3^. In this case, the DNA molecule is placed in the through channels of the cristal near the trimers at a distance sufficient for all three N-terminal lysines (Lys5, Lys8, Lys10) of the protein to participate in DNA binding.

The importance of the results of this work, which brings researchers closer to completely deciphering the structure of the intracellular DNA–Dps crystal, is not only scientific. The discovery of this phenomenon could be of great practical importance, as the Dps protein is increasingly being considered as a potential drug carrier, especially for anti-tumor drugs. The discovery that the oligomeric structure of the Dps protein changes from dodecameric to trimeric upon interaction with DNA opens up new perspectives in this research.

## Figures and Tables

**Figure 1 ijms-26-00619-f001:**
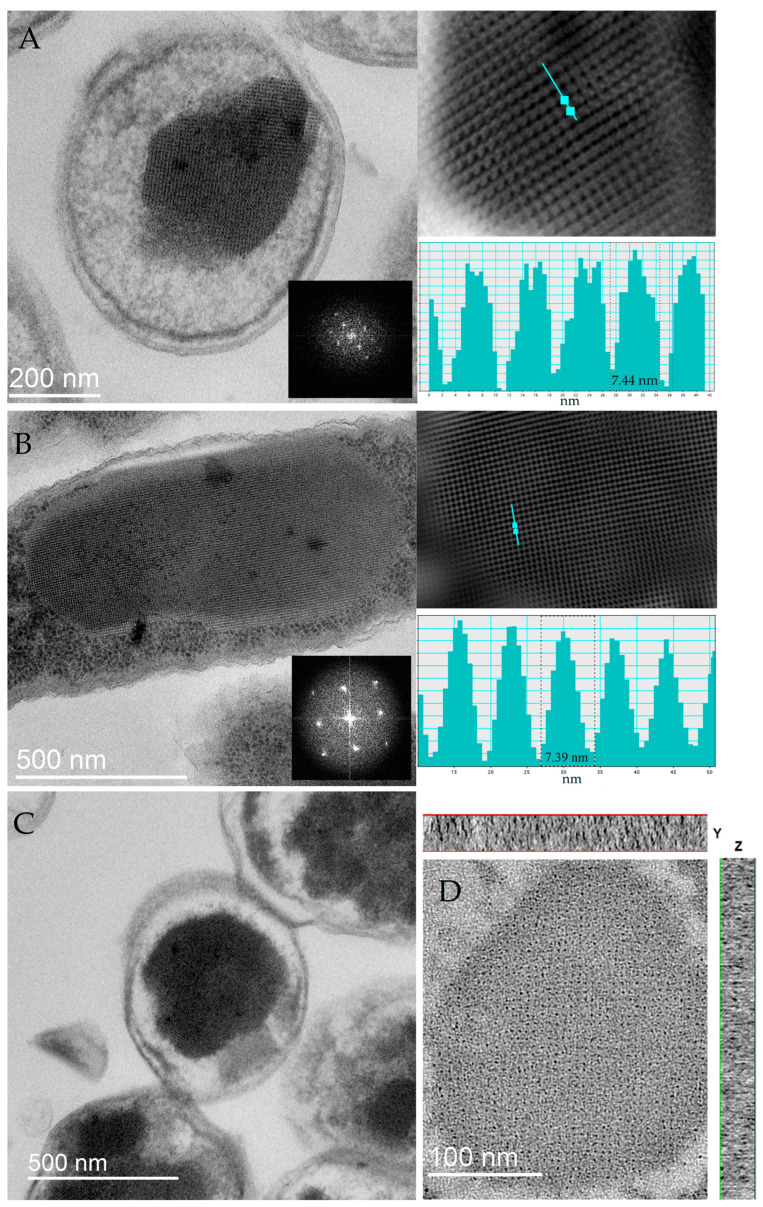
Electron micrographs of 2-day-old stationary *E. coli* Top cells: (**A**) TEM image of a typical cell with a small crystal, obtained in the absence of Dps protein overexpression. The inset on the lower right is the Fourier transform of the region containing the crystal. Top right—Fourier filtered image of the cell crystal, bottom right—intensity plot along the turquoise line, the dashed line indicates a repeat element of ~7.44 nm. (**B**) TEM image of a typical cell with a large crystal (cross-section) obtained under Dps protein overexpression conditions. The inset on the lower right is the Fourier transform of the region containing the crystal. Top right—Fourier filtered image of the cell crystal, bottom right—intensity plot along the turquoise line, the dashed line indicates a repeat element of ~7.39 nm. (**C**) TEM image of a typical cell with a large crystal (longitudinal section) obtained under conditions of Dps protein overexpression. The inset in the lower right corner is the Fourier transform of the region containing the crystal. (**D**) Tomogram of a cell containing a two-dimensional intracellular crystal. The slice thickness is 100 nm. The rectangles at the top and right show the intensity along the corresponding axes along the thickness of the slice.

**Figure 2 ijms-26-00619-f002:**
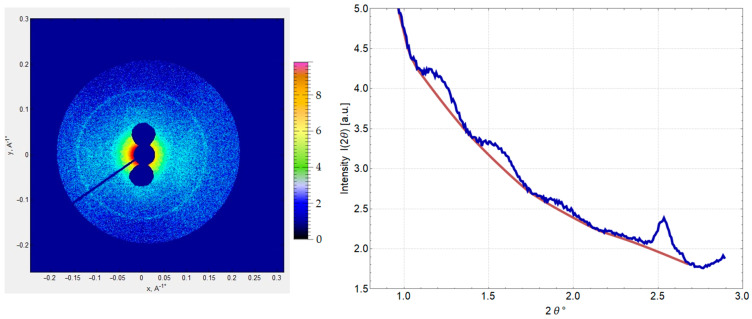
SAXS results of 2-day stationary *E. coli* Top cells with an intracellular DNA–Dps crystal. (**A**) Two-dimensional distribution of the intensity of cell samples. (**B**) One-dimensional curve of the intensity of X-ray scattering from intracellular DNA–Dps crystals depending on the scattering angle 2θ (blue). Scattering intensity by amorphous cell structures that are not related to intracellular crystals (red). The scattering magnitude as a function of 2θ was estimated in Wolfram Mathematica with the Estimated Background function.

**Figure 3 ijms-26-00619-f003:**
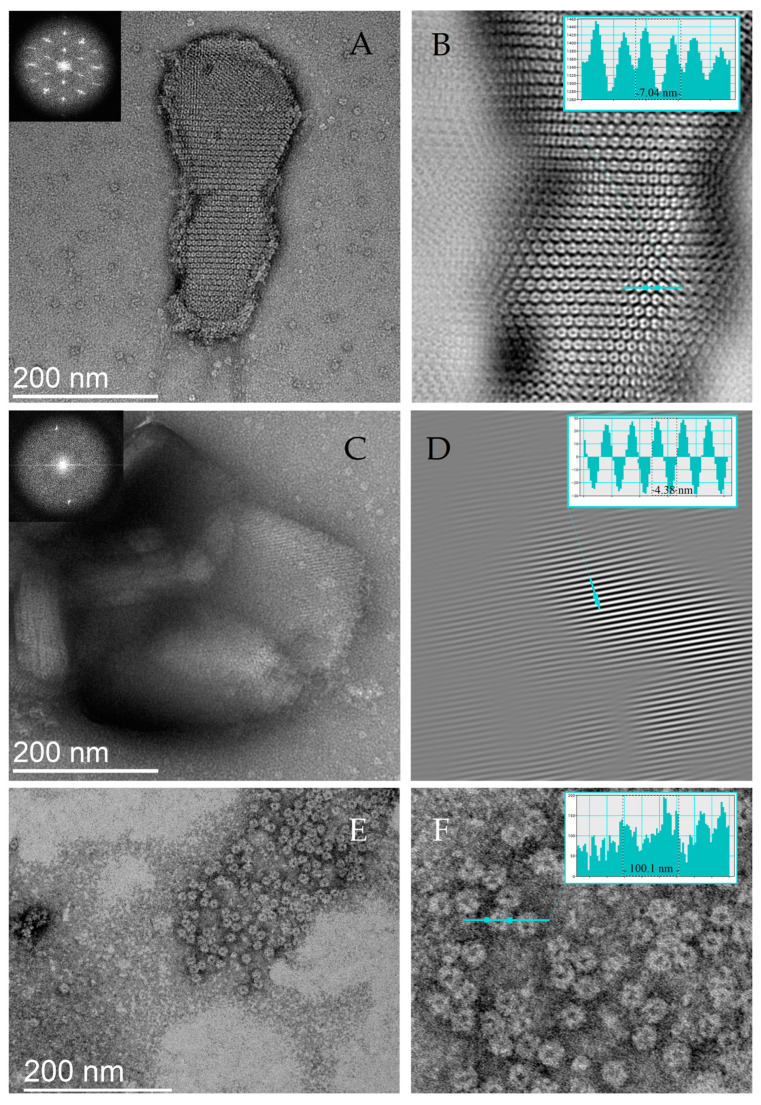
Images of in vitro DNA–Dps crystals. (**A**) TEM image of the crystal in the frontal projection. The inset in the upper left corner is the Fourier transform of the image. (**B**) Fourier filtered image of (**A**). The inset shows the intensity along the turquoise line; the dashed rectangle shows the size of the repeating element of ~7 nm. (**C**) TEM image of the crystal in the lateral orientation. The inset in the upper left corner is the Fourier transform of the region highlighted in the black box. (**D**) Fourier filtered image of (**C**). The crystal is in an orientation where periodicity is observed in only one direction (interlayer distance ~4.4 nm), different from the plane in the image in (**A**). (**E**,**F**) TEM images of Dps protein in solution. The formation of periodic structures is not observed; the particle size is ~9 nm.

**Figure 4 ijms-26-00619-f004:**
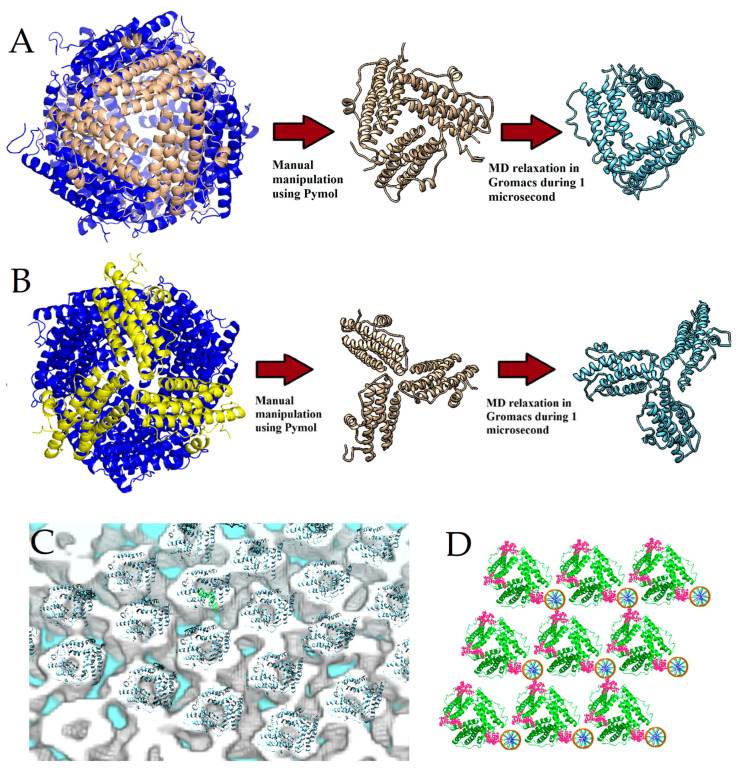
Modeling of the trimer structure from the dodecameric structure of the Dps protein using molecular dynamics methods. The protein is shown in blue. The three pore-forming subunits are highlighted in a different color. Trimers of two types were simulated for 1 μs in Na^+^Cl^−^ solution. The initial positions of the monomers were obtained by cutting out (**A**) the ferritin pore (ΔΔG_Ferr_ = −185 kJ/mol) and (**B**) the Dps-type pore (ΔΔG_Dps_ = −115 kJ/mol). The red arrows show the structure at the initial time (tan) and after the simulation (blue). (**C**) Arrangement of the trimers simulated by molecular dynamics methods into the electron density tomography. Shown are electron density (grey-white) and a superimposed protein models (blue helices). (**D**) Arrangement of DNA molecules into the crystal structure of DNA–Dps. Protein trimers are shown in green. The ten N-terminal amino acid residues of Dps (including the main DNA-binding amino acid residues Lys5, Lys8, and Lys10) are marked in pink. The orange ones that look like wheels are DNA in cross-section.

**Figure 5 ijms-26-00619-f005:**
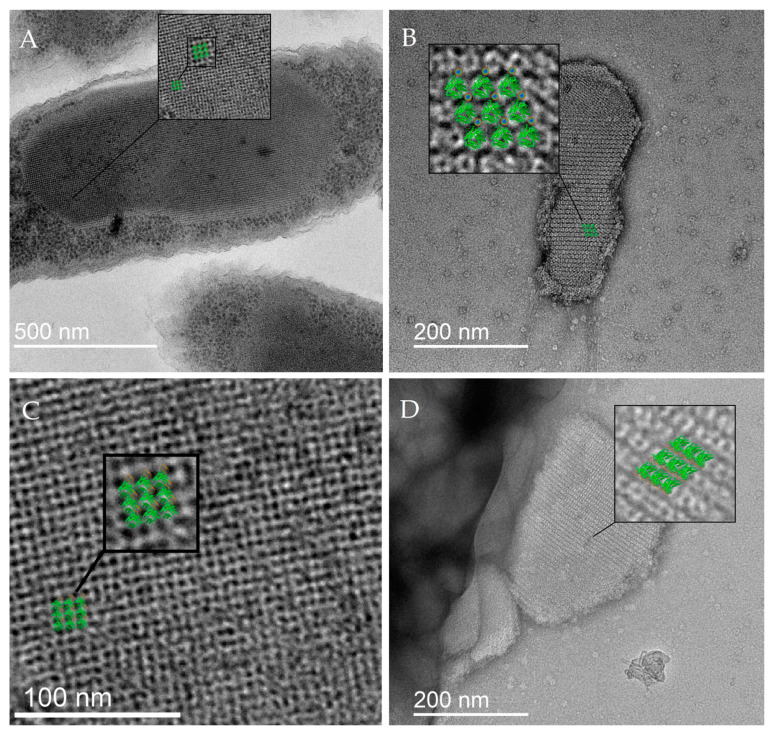
Superposition of the crystal model obtained from in vitro tomography data of DNA–Dps crystals with TEM images of intracellular crystals of 2-day-old *E. coli* Top cells: (**A**–**C**)—frontal orientation; (**D**)—lateral orientation.

**Figure 6 ijms-26-00619-f006:**
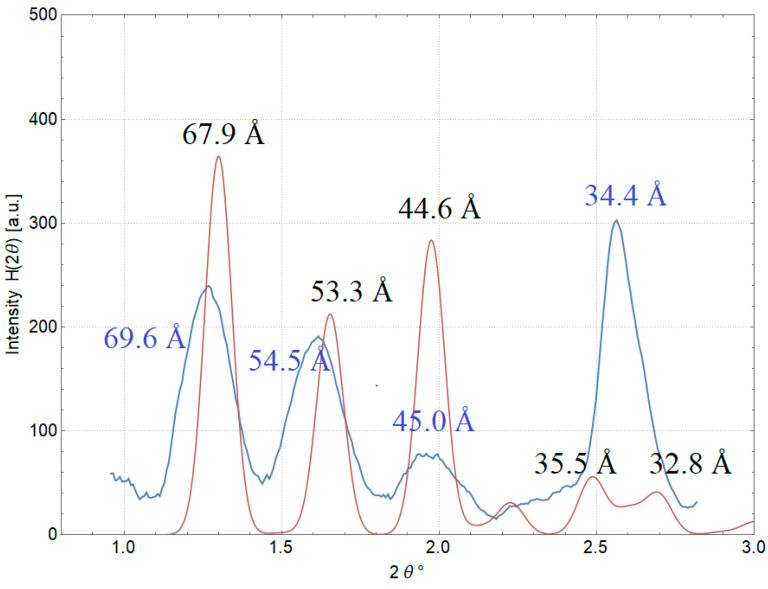
Comparison of the experimental one-dimensional scattering curve from intracellular crystals of 2-day stationary *E. coli* Top cells (blue) and the scattering curve (red) from a simulated crystal of DNA–Dps with the protein in trimeric form.

**Figure 7 ijms-26-00619-f007:**
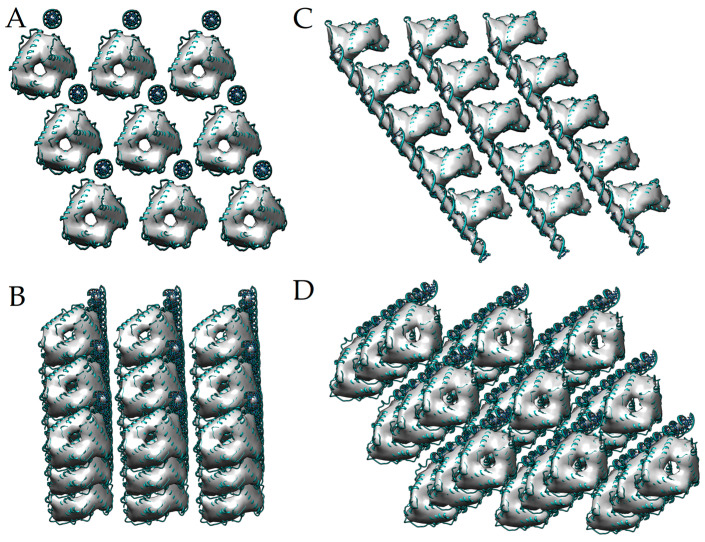
The structure of the DNA–Dps crystal obtained by molecular modeling methods. (**A**) front view; (**B**,**C**) side view; (**D**) view of the crystal in the tomography plane.

**Figure 8 ijms-26-00619-f008:**
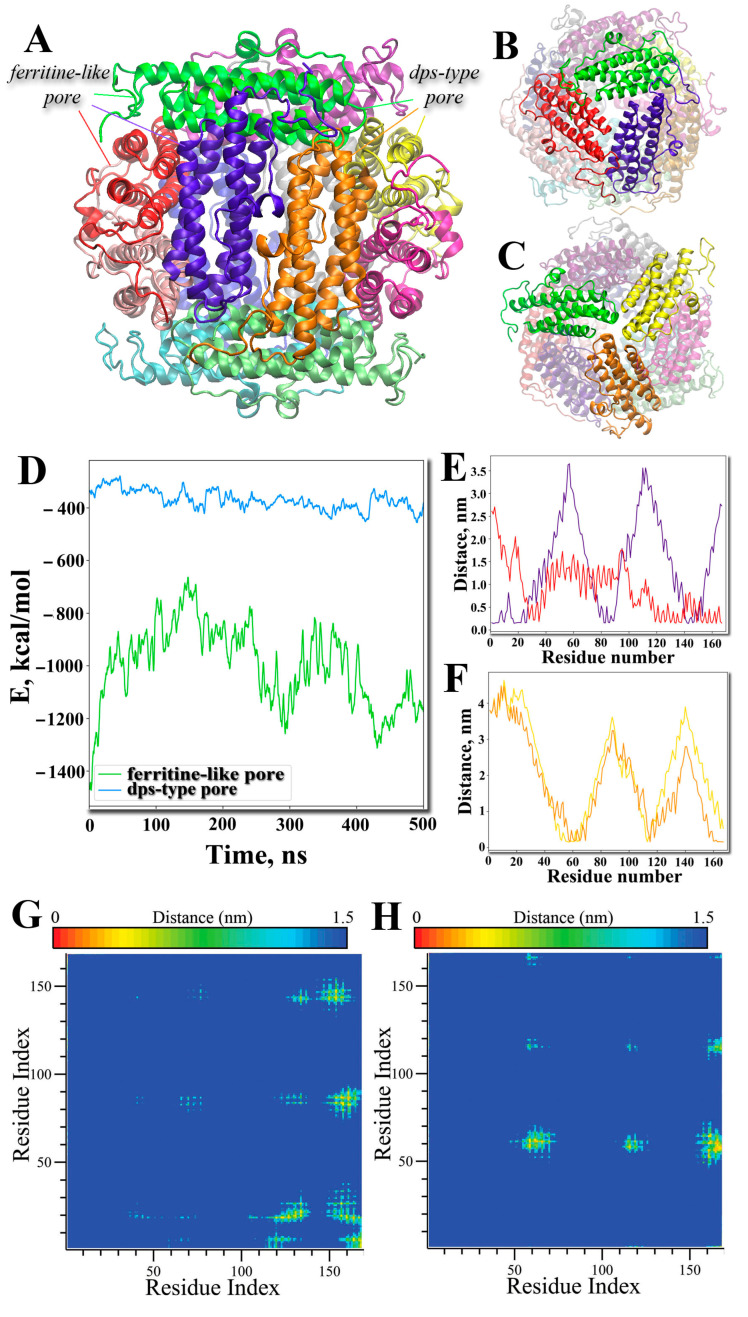
The structure of the Dps dodecamer (side view) in which each subunit is shown in its own color (**A**). Of the four ferritin-like pores and four Dps-type pores, examples are shown in one of the pores. The lines correspond to the colors of the subunits that form the pore. Ferritine-like pore view. (**B**), Dps-type pore view. (**C**). Different subunits are shown in different colours. The pore-forming subunits are highlighted brightly. Potential energy of interaction of subunits (**D**) within a ferritin-like pore (green) and a Dps-type pore (ice blue). Minimum distances between a green-colored residue and the other two residues in a ferritin-like pore (**E**) and a Dps-like pore (**F**). The colors of the curves in (**E**,**F**) correspond to the colors of the subunits with which the green-colored subunit interacts in (**B**,**C**). Mean smallest distances (contact map) between two subunits interacting within a ferritin-like pore (**G**) and a Dps-type pore (**H**). The color transition in the figures corresponds to distances from 0 nm (red color, close distance) to 1.5 nm (blue color, no interaction), see scales above the figures.

**Figure 9 ijms-26-00619-f009:**
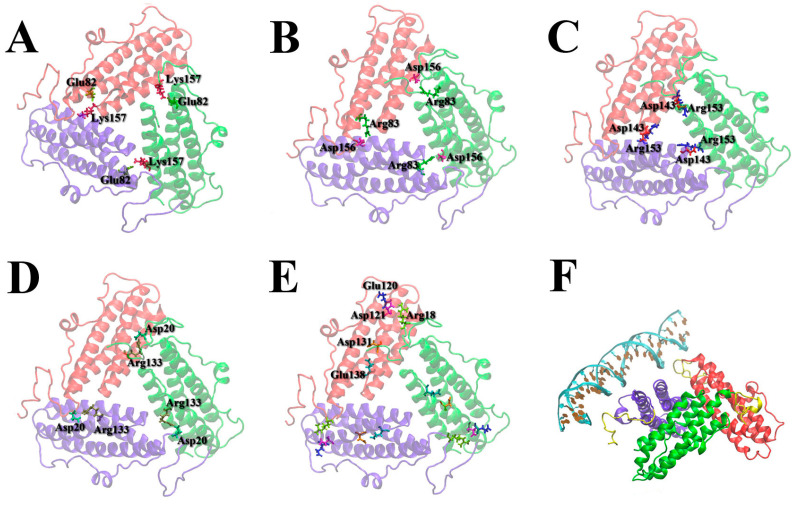
The main interacting amino acid residues that determine the binding of ferritin-like trimers: Glu82-Lys157 (**A**), Arg83-Asp156 (**B**), Asp143-Arg153 (**C**), Asp20-Arg133, Arg18 (of flexible N-terminus)—Glu120, Asp123, Asp131, Glu138 (**E**). The key amino acid residues involved in subunit binding within the trimer are indicated by three-letter code and number (**A**–**E**). Side view of a ferritin-like Dps trimer binding DNA (**F**). DNA is shown in ice blue. The flexible N-termini (amino acid residues 1–20) are shown in yellow in (**F**). The three interacting subunits of the Dps protein are shown in red, green, and purple in (**A**–**F**).

## Data Availability

Data is contained within the article.

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
