# Peer review of "The Dps Protein Protects *Escherichia coli* DNA in the Form of the Trimer"

_ijms, 2025, doi:10.3390/ijms26020619_

Round 1

Reviewer 1 Report

Comments and Suggestions for Authors

The reviewed article investigates the Dps protein of Escherichia coli, focusing on its role in protecting DNA under stress conditions through the formation of crystalline structures. The authors propose that the Dps protein transitions from a dodecameric to a trimeric oligomeric state, enabling DNA encapsulation in a toroidal structure. This conclusion is supported by a combination of in vitro and computational studies, including molecular dynamics and electron microscopy. The findings contribute to understanding bacterial DNA protection and have potential biotechnological and medical applications.

Comments

1. The transition of the Dps protein from dodecamer to trimer is a key claim. However, the evidence relies heavily on computational modeling and TEM images, with limited direct biochemical or structural validation. To confirm the presence of trimers, additional experimental verification of the trimeric form, such as cryo-EM or cross-linking experiments, as well as biochemistry assays, should be included. 

2. The study mentions factors like sugar metabolites influencing Dps oligomerization but does not explore these in detail. The role of intracellular conditions such as pH and ion concentration is also underdeveloped.  The work would benefit if the authors perform controlled in vitro experiments to systematically investigate the conditions favoring the trimeric state and link these findings to intracellular environments.

3. The TEM images are derived from overexpressed Dps protein, which may not accurately represent physiological conditions given potential overexpression artifacts could affect the crystal structure.  Analyses from wild-type strains could help to ensure that the findings are not influenced by overexpression.

4. While the structural transitions are well-characterized, the functional implications for DNA protection and bacterial survival are not experimentally demonstrated. The authors suggest that the trimeric form enhances DNA protection compared to the dodecameric form under stress conditions. Further in vitro are needed to assess your discussion as the biological significance of the trimeric state remains speculative. The transition's role in bacterial survival under specific stress conditions is not directly tested. 

5. The discussion about the biological relevance results is insufficient for the scope of the work. To expand on the relevance of the trimeric form by performing stress assays to determine its advantage in oxidative or nutrient-deprived environments could increase the clearness of the work.

6. The molecular dynamics simulations rely on specific force fields and assumptions about protein flexibility and environmental parameters. These choices could bias the results.  The authors hypothesize mechanisms for oligomeric transition, but the details remain unclear. For example, how specific residues or regions of the Dps protein contribute to this process is not explored.  The authors should validate the computational model with experimental data, like mutagenesis studies, which would confer to the study a higher interest and quality.

Addressing these limitations will greatly enhance the reliability and impact of the study.

Reviewer 2 Report

Comments and Suggestions for Authors

In the abstract, please remove the term first time.

Why do authors specifically study Escherichia coli bacteria? Any reason?

Line 43, why underline to D, P, etc.,

Line 89, in vitro term should italics check another place.

What is the limitation of this study?

Compare your results with previous studies.

Give proper mechanism from a chemist's point of view.

Comments on the Quality of English Language

Need to improve

Round 2

Reviewer 2 Report

Comments and Suggestions for Authors

The manuscript is improved. However, English has many problems which are not readable to readers. For ex, In abstract line 23, the authors wrote, We prove that, "what we prove" needs to be mentioned after we need to write how, and so on...

Therefore manuscript need extensive English revison. 

Author Response

Dear reviewer. We have taken your comment into account and corrected the text to make the English language of the article more correct and understandable. Thank you.